# Experimental Correlation of the Role of Synthesized Biochar on Thermal, Morphological, and Crystalline Properties of Coagulation Processed Poly(1,4-phenylene sulfide) Nanocomposites

**DOI:** 10.3390/polym15081851

**Published:** 2023-04-12

**Authors:** Zaib Un Nisa, Lee Kean Chuan, Beh Hoe Guan, Faiz Ahmad, Saba Ayub

**Affiliations:** 1Department of Fundamental and Applied Sciences, Universiti Teknologi PETRONAS, Seri Iskandar 32610, Perak, Malaysia; 2Department of Mechanical Engineering, Universiti Teknologi PETRONAS, Seri Iskandar 32610, Perak, Malaysia

**Keywords:** synthesis, characterization, carbonization, coagulation, bio char, nanocarbon, morphological

## Abstract

This work aimed to study the thermal and crystalline properties of poly (1,4-phenylene sulfide)@carbon char nanocomposites. Coagulation-processed nanocomposites of polyphenylene sulfide were prepared using the synthesized mesoporous nanocarbon of coconut shells as reinforcement. The mesoporous reinforcement was synthesized using a facile carbonization method. The investigation of the properties of nanocarbon was completed using SAP, XRD, and FESEM analysis. The research was further propagated via the synthesis of nanocomposites through the addition of characterized nanofiller into poly (1,4-phenylene sulfide) at five different combinations. The coagulation method was utilized for the nanocomposite formation. The obtained nanocomposite was analyzed using FTIR, TGA, DSC, and FESEM analysis. The BET surface area and average pore volume of the bio-carbon prepared from coconut shell residue were calculated to be 1517 m^2^/g and 2.51 nm, respectively. The addition of nanocarbon to poly (1,4-phenylene sulfide) led to an increase in thermal stability and crystallinity up to 6% loading of the filler. The lowest glass transition temperature was achieved at 6% doping of the filler into the polymer matrix. It was established that the thermal, morphological, and crystalline properties were tailored by synthesizing their nanocomposites with the mesoporous bio-nanocarbon obtained from coconut shells. There is a decline in the glass transition temperature from 126 °C to 117 °C using 6% filler. The measured crystallinity was decreased continuously, with the mixing of the filler exhibiting the incorporation of flexibility in the polymer. So, the loading of the filler into poly (1,4-phenylene sulfide) can be optimized to enhance its thermoplastic properties for surface applications.

## 1. Introduction

The carbon products obtained from industrial sources have been used as fillers in various polymers [1,2,3]. This has been extensively practiced in industrial sectors to advance the routine properties of polymeric products [4,5,6]. There have been dynamic research within industrial and academic fields to study the composites and nanocomposites of polymers. Polymeric composites have proven to be a major and unremarkable endeavor. The major technical and commercial interest lies in the ubiquitous existence of properties in consumer products. Polymers modified with carbon products have frequently been investigated for their thermal, conductive, thermoelectric, tensile, mechanical, corrosion, and erosive properties [7,8,9,10].

The production of biochar (BC) from agricultural waste is the leading and cheapest source for the production of nanocarbon as an industrial product, with a predicted yearly production of fifteen metric tons by 2025 [11]. BC is fundamentally produced from the carbonization of carbon resources, such as biomass, fossil fuels, and biofuels [12]. Thus, to achieve large-scale productivity, low-cost raw material, and inexpensive products with the required properties, BC can be seen as an appealing choice [13,14].

It is a known fact that carbon-based materials, such as carbon black, chars, or activated carbon, have demonstrated finer pore size development than wood-based carbon products. The majority of the pore volume has shown a radius of <1 nm in the case of coconut shells, while carbon material originating from wood has a significant number of macropores and mesopores [15,16]. A survey was performed on various agricultural by-products, such as almond shells and seeds of peaches, grapes, cherries, apricots, and palms, with the conclusion that the botanical origin of the family of the selected material also affects the distribution of pore size [17,18]. BC specifically obtained from coconut shells as a bio-carbon source has the potential for substituting the conventional thermoset filler in wear applications due to their high strength, low density, hardness, abrasion resistance, and modulus properties [19,20].

The tailoring of inbuilt characteristics and the achievement of the required characteristics of a matrix is usually accomplished by using fillers. Poly (1,4-phenylene sulfide) (PPS) is the most extensively utilized semicrystalline polymer. It retains superior thermal, chemical, mechanical, anti-aging, flame-resistant, water-resistant, low thermal expansion coefficient, and exceptional friction properties [21,22]. Surface protection is one of the major requirements of industries using high-temperature conditions. The PPS has been successfully employed as a protective coating because of superior melt temperature, good chemicals, and excellent abrasive resistance. PPS is a high-performance thermoplastic, so its high-performance thermoplastic composites are taking a progressively significant role in thermoplastic applications. The benefits include the properties, such as elevated impact resistance, and toughness; better chemical and corrosion endurance; easy processing; indefinite shelf lifespan of the prepregs; and the ability to be recycled [23,24].

In the category of carbon-based composites of PPS investigation has been carried out using glass fiber (GF), carbon fiber (CF), metal oxides, graphene, Mxene, short carbon fibers (SCFs), nanodiamonds (NDs), and graphene oxide (GO). The systematic evaluation of behavioral isothermal and non-isothermal crystallization of PPS modified with short glass fiber has been conducted using different models. The details of the crystallization evolution, dynamic mechanical properties, and aging effects using heat treatment are available [25]. The PPS and CF-reinforced composites have been investigated for thermomechanical, fracture-resistant, and tribological properties [26]. Short carbon fibers of graphene oxide and nanodiamond (ND)-based composites of PPS have been reported for their ability to be utilized in membranes, friction reduction, and wear applications [27]. Its conductive polymer composites have been discovered to be feasible replacements for the electrolytic membranes of fuel cells and bipolar plates [28].

Research findings on the use of activated nanocarbon in various types of material manufacturing are still in development because of its abilities in renewable energy harvesting [29,30] and environmental applications—especially focusing on global challenges in clean energy, sensors [31], dielectric and EMI applications [32], high-performance batteries [33], electrochemical energy storage, and environmental remediation [34,35]. The addition of BC to elastomeric materials has been studied in many industrial applications [36,37]. In current research observing the easy-to-tailor and modifiable properties of PPS for many useful applications, its undiscovered biochar-based composites (PPS@BC) were synthesized. BC was synthesized from raw coconut shells. As a high-performance engineering polymer, the thermal and crystallization properties of PPS are shown to be affected by the percentage of blending, technique of synthesis, and nature of reinforcement [29,38].

Herein, a study was conducted to explore the thermal, morphological, and crystalline properties of PPS@BC composites. The composites were made using the coagulation protocol. The investigation was conducted by loading different percentages of synthesized BC into the PPS matrix, and the variations in the properties were studied. 

## 2. Experimental

### 2.1. Raw Materials

PPS powder (average Mn = 10,000 g mol^−1^, density = 1.36 g/mL at 25 °C) was supplied by Sigma-Aldrich (St. Louis, Missouri, USA); an NMP reagent grade was purchased from Supelco (Darmstadt, Germany, b.p. = 202 °C). Methanol was obtained from Friendemann Schmidt (sdn bhd, Kuala Lumpur, Malaysia, b.p. = 64.9 °C) and used as received. The mesoporous biochar (BC) was synthesized from the coconut shell and characterized in the laboratory using the following carbonization technique. The schematic diagram for the workflow is presented in Figure 1. 

### 2.2. Synthesis of Biochar BC

After being washed with distilled water, coconut shells were dried at 80 °C in an oven. The dried shells were granulated using a granulator to obtain smaller grains. The prepared grains of the coconut shells were treated in a tube furnace to bring out the carbonization as follows [39]. 

The ceramic boat was loaded with grains of coconut shells and treated thermally in a tube furnace (PROTHERM, France, model: PTF 12/75/800, tube type: C610) that had the dimensions of DxWxH 635 × 850 × 400. The carbonization was carried out under nitrogen at 700 °C with a 10 °C/min increase in temperature. Following thermal treatment for two hours, the carbonized product was allowed to cool to room temperature, with an annealing time of eight hours. The thermally degraded product was further ground in dry form using a laboratory blender (WARING, McConnellsburg, PA, USA, model No. HGBTWTG4) to obtain a powder. The obtained BC was further sieved with 3 U.S. mesh to obtain a fine powder. 

### 2.3. Physical Activation of BC

The synthesized biochar was physically activated at a temperature of 700 °C in an atmosphere of carbon dioxide. The flow of carbon dioxide was set to 100 mL/min. The heating temperature and gas flow remained constant for six hours. After the physical activation, the product was brought to room temperature under nitrogen [40].

### 2.4. Synthesis of PPS@BC Nanocomposites Using the Coagulation Method 

PPS was dissolved in N-Methyl-2-pyrrolidone (NMP) at 350 °C using a hot plate until a clear solution was obtained. Synthesized mesoporous BC was evenly dispersed as filler in NMP using ultrasonication for 30 min in a separate beaker. The two solutions were mixed with five different compositions of 2%, 4%, 6%, 8%, and 10% of mesoporous nanofiller. The homogenous dispersion of BC was achieved through ultrasonication of the mixture for 30 min. Then, 50 mL of methanol was added to the reaction mixture with continuous stirring to obtain the flocculates. The reaction mixture was further ultrasonicated for another 10 min. After the filtration of the coagulated nanocomposite, the product was dried at 80 °C under a vacuum.

## 3. Characterization Techniques

### 3.1. Characterization of BC 

The characterization techniques utilized to study the morphology, elemental identification, phase composition, crystallinity, particle size, dimension, and specific surface area of the BC were BET, XRD, and SEM with EDS.

Brunauer, Emmet, and Teller (BET) evaluations were made using a Micromeritics (ASAP 2020) instrument, and the specific surface area of the product was measured through N_2_ adsorption. The analysis of the X-ray powder diffraction (XRD) instrument was performed on (Bruker, Karlsruhe, Germany, AXS, D8 advance) using Cu Kα (λ_1_ = 1.54056 °A and λ_2_ = 1.5444 °A, with a ratio of λ_2_/λ_1_ = 0.5) and a radiation source (45 kV and 40 mA) in continuous scanning mode. The samples were examined at a scanning speed of 1° min^−1^ from 10° to 100° on a 2θ scale. The Carl Zeiss (Oberkochen, Germany, Supra 55vp) instrument was utilized to achieve field-emission scanning electron microscope (FESEM) images.

### 3.2. Characterization of PPS@BC Composites

Four characterization techniques were utilized to analyze the structural, thermal, and morphological properties of PPS@BC nanocomposites. The Fourier transform infrared spectra were recorded on an FTIR instrument (Perkin Elmer; Shelton, CT, USA, FTIR Frontier) using KBr solid-state analysis. Thermal characterization was carried out using a Perkin Elmer (STA6000) thermobalance at a heating rate of 10 °C/min under an N_2_ atmosphere of up to a maximum temperature of 600 °C. The differential scanning calorimetry (DSC) analysis was performed under N_2_ using a Perkin Elmer (Pyris-1) instrument. The samples were heated from 50–800 °C for a complete cycle at a heating rate of 10 °C/min. The field-emission scanning electron microscope (FESEM) imaging was obtained using a Carl Zeiss (Supra 55vp) instrument.

## 4. Results and Discussions

### 4.1. Surface Morphology and Composition of BC

Field emission scanning electron microscopy (FESEM) imaging with energy-dispersive X-ray spectroscopy (EDS), is a valuable analytical tool for surface study in terms of morphology and composition using EDS. The observed images of the synthesized nanocarbon at different magnifications, captured at an acceleration potential of 5 to 200 kV, are presented in Figure 2. The obtained images exposed the porous morphology of the carbonized coconut shell product. This acknowledges the fact that the carbonized product of the coconut shell, at given conditions, is feasible for developing nanocarbon with a better surface area, which is later supported by the SAP results. Therefore, the carbonization temperature and time, for the physical activation of materials, were proven to play crucial roles in the carbonization process.

The backscattered electron (BSE) image and composition, as determined by weight using EDS analysis, is presented in Figure 3. The prepared BC contains 90.2% carbon, 9.5% oxygen, and 0.3% potassium by weight fraction, as listed in Table 1. Therefore, most of the ingredients of the coconut shell were well-degraded, and carbonized coconut shell residue, in the form of the required biochar, contained a copious amount of carbon.

### 4.2. SAP Analysis of BC 

Parameters related to the surface of a material can be used for physical adsorption phenomena. The Brunauer–Emmett–Teller (BET) theory is used to explain the physical adsorption of gas molecules on a solid surface. This analytical technique provides useful information for the measurement of the specific surface area of materials. The underlying phenomenon is the physical adsorption of gas molecules on the surface of the material via London dispersion forces. The International Union of Pure and Applied Chemistry (IUPAC) has recognized that absorbent surfaces can give rise to five conventional types of isotherms [31]. The SAP study of the product presented smooth type III isotherms according to the IUPAC classification, thus predicting stronger adsorbent–adsorbate interactions.

The adsorbate dissemination into a material was utilized in SAP analysis. The volume, size, and diameter of the pores are directly related to the adsorption capacity of a surface. This selectivity, in an appropriate structure with requisite pore considerations, is of key interest for a particular application [41]. The selection of raw material, the processing conditions, and an adequate activation surface play key roles in pore development [42,43].

The achieved isotherm linear curve, including the size distribution analysis of BC at the STP, is provided in Figure 4. The initial part of the linear isotherm plot of the isotherm for BC represents the saturation of the pores, with nitrogen gas used for analysis. This is shown as the BC adsorption curve (ad). The second curve, De, is abbreviated for the desorption analysis of BC. The slope of the plateau at relatively high pressure depicts the multilayer adsorption of the absorbate on the porous BC surface [36]. The details of the measured SAP parameters are provided in Table 2. The average pore diameter was calculated to be of nano size (2.51 nm) with specific surface area (SBET) and Langmuir surface (SL) area values of 1517 m^2^/g and 2175 m^2^/g, respectively. In the literature, char materials are usually reported with chemical activation. In a review of the chemical activation of agricultural residues, the SBET was reported with a range of 0.1–1718 m^2^/g [44]. In the current study, the area achieved was comparable to the maximum achieved value, even when using facile processing parameters.

### 4.3. X-ray Diffraction Analysis of BC

X-ray diffraction analysis (XRD) is known as a useful, nondestructive technique used to obtain comprehensive knowledge about the crystal structure, chemical composition, crystallographic parameters, and physical properties of a material [44,45]. Carbon is made up of a combination of heterogeneous atoms, and their thickness can vary from a monolayer to a multilayer arrangement. So, there is a small change in crystallographic parameters depending on the processing parameters and raw material. The obtained XRD spectra for BC are provided in Figure 5. The reported structural parameters of BC, in terms of the interplanar spacing (d_hkl_), full-width half maximum (FWHM), the crystallite size (L_c_), lateral size (L_a_), No. of layers (N), dislocation density (δ), microstrain (ε), packing density (ρ), and crystallinity index, were calculated using the method from the literature [46]. The XRD spectral pattern revealed two strong Braggs peaks associated with reflections from the (002) and (10) planes. The calculated peak position of BC on the 2θ scale, in degrees, is presented in Table 3. The major peak, due to reflection from the (002) plane, is located at 24.8° on the 2θ scale. The relatively smaller peak, at 40.7° on the 2θ scale, is a result of two-dimensional reflection from the (10) plane. The calculated microstructural parameters for the synthesized BC are tabulated in Table 4. It is supported by the reported results that the observed microstructural parameters show small lattice dimensions and degrees of graphitization [47,48,49]. The percent crystallinity of BC was calculated using the following Formula (1).
(1)Crystallinity Index (%)=(I crystalline/I total)×100

The intensity ratio of the crystalline peak was related to the amorphous peak, using their respective mass ratios, and the degree of crystallinity was accomplished [46]. A calculated crystallinity index value of 53.07% was observed for BC. 

### 4.4. FTIR Interpretation of PPS@BC Nanocomposites

PPS@BC nanocomposites were synthesized using the coagulation technique. The coagulant used was methanol, and the neat polymer was also analyzed along with synthesized nanocomposites. The samples were investigated from 4000–550 cm^−1^, and the results are provided in Table 5. The structure of PPS most likely resembles para-substituted benzene; however, it is challenging to assign the individual modes.

The obtained spectra are presented in Figure 6. In the spectra, the highly noticeable peaks are due to symmetric and asymmetric benzene ring stretch [50]. These appear in the region from 1300 cm^−1^ to 1600 cm^−1^. The peak assigned, due to the symmetric stretch, was sharp compared with the wide peak of the asymmetric stretch. These peaks were shifted to higher frequency values in the case of nanocomposites compared with neat polymers. Aromatic C–H stretching vibrations are not very prominent and showed a small, broad peak in the region of 2300 cm^−1^. 

The carbon atom of the substituted benzene ring linked with sulfur reflected a wide band above 1800 cm^−1^. The out-of-plane C–H vibrations presented distinctive medium peaks in the region of 807–811 cm^−1^ [51]. The characteristic C–S stretching vibrations of the benzene ring were present in lower frequency regions above 450 cm^−1^. Variations in the band intensity were noticed as synthesized filler was added into the neat PPS polymer. Such changes may be due to changes in the symmetry and crystallinity of the matrix upon the addition of synthesized nanocarbon.

The characteristic C–S aliphatic stretching vibrations of the benzene ring were assumed to be present from 1084 cm^−1^. to 1089 cm^−1^. This aliphatic C–S bond is present at a higher frequency than the C–S bond due to the benzene ring. This trend is obvious due to the restricted movement of sulfur attached to the carbon of the benzene ring. The electronic cloud of the diffused benzene ring has lowered the movement of the adjacent bulky sulfur atoms of the polymer backbone. There was a noticeable peak at 3433 cm^−1^, which is due to the formation of an additional O–H bond. This bond was absent in the neat polymer. It was assumed that these O–H bonds were created during the coagulation process, where methanol was used as a coagulant. During the sonication of methanol with the polymer skeleton, a productive interaction occurred that created polar bonds on the PPS matrix. The intensity of this bond increased with the increase in filler content, up to an additional 4%, after this intensity of the O–H bond was decreased again. This indicated that an extra quantity of filler hindered the beneficial interaction of methanol with the polymer skeleton. This increase in the polar character of the polymer matrix incorporated strong intermolecular forces, which can be beneficial for improving the mechanical, thermal, and surface-related properties of the synthesized nanocomposite. 

### 4.5. Thermogravimetric Analysis of PPS@BC Nanocomposite

Thermograms of neat PPS and PPS@BC nanocomposites are presented in Figure 7. The details for the data are listed in Table 6. The thermal stability of the nanocomposites was predicted using the information about the temperature at 5% weight loss (T_5_), the temperature at 10% weight loss (T_10_), the temperature at 50% weight loss (T_50_), the final degradation temperature (T_f_), and the residual weight at T_f_ (R_f_). All the PPS@BC nanocomposites, along with the neat PPS, were degraded above 640 °C. The addition of 6% BC resulted in a rise of 13 °C in T_f_ compared with neat polymer. This suggests that the synthesized nanocomposite can be utilized at a somewhat higher temperature. Overall, 5%,10%, and maximum weight loss temperatures were found to be highest for PPS-6, suggesting that its initial weight loss would take place at a higher temperature. 

The increase in thermal stability from 90 °C to 157 °C was noticed while raising the filler content from 2 to 6%. Half weight loss temperature of the nanocomposites ranged from 520–580 °C for all the prepared nanocomposites. It was evident from thermograms that, for the 2 to 4 percent loading of the filler into the PPS, the nanocomposites showed comparable thermograms. For the nanocomposites with 8 and 10 percent filler contents, there was a drastic decline in T_5_, T_10_, and T_50_. This indicates that, with greater loading of the filler, initial weight loss will take place at low temperatures. This may be due to the easy residual solvent loss for composites with higher filler content. The residual weight (R_f_) was calculated to be the same for 2 to 6% filler content, and the lowest residual weight was observed for the 10% loading of nanocarbon. It was concluded that mixing 6% of the synthesized nanocarbon obtained from the coconut shell into PPS provided the optimum concentration that could be utilized. Its final degradation happened at 13 °C higher than neat PPS, which suggests that the presence of nanocarbon prepared from coconut shells delayed the oxidative degradation of PPS. A comparative illustration of thermal stability for all of the samples is presented in Figure 8.

### 4.6. Differential Scanning Calorimetric Investigation of PPS@BC Nanocomposites

The literature reveals that the modification of thermoplastic material results in variations in crystallinity behavior [50]. These crystallinity changes can be utilized to study the effects on mechanical characteristics. DSC analysis provides an indirect path to calculate the degree of crystallinity and glass transition temperature of polymer composites. DSC curves of the PPS and coagulation-processed PPS/BC nanocomposite were studied over a temperature range from −50 °C to 350 °C. The DSC results for melting and cooling cycles are displayed in Figure 9. All samples showed an endothermic melting peak, with variations in the values of the cooling temperature (T_c_), melting temperature (T_m_), heat developed during cold crystallization (∆H_c_), heat absorbed during melting (∆H_m_), degree of supercooling ∆T (T_m_–T_c_), degree of crystallinity (X_c_), and glass transition temperature (T_g_). All the thermal parameters for crystallization and melting behavior are listed in Table 7. The results are tabulated from the heating–cooling–heating scans of PPS@BC nanocomposites. The degree of crystallinity was calculated from Equation (2) [51]:(2)$$\%\;X_{c}=\frac{∆H_{c}}{∆H_{f}-(1-W_{f})}\times 100$$
where X_c_ = degree of crystallinity,

∆H_c_ = heat of crystallization,

∆H_f_ = heat of crystallization for a 100% crystalline PPS, 112 J/g [52], and

W_f_ = weight fraction of BC content in the nanocomposite.

It was revealed that adding BC nanofiller increased the onset temperature of crystallization (T_oc_) and the crystallization temperature (T_c_) of PPS@BC nanocomposites compared to neat PPS. With the gradual addition of filler, the maximum attainable value of T_c_ was 258 °C for the selected series of the nanocomposite. The onset temperature of melting (T_om_) and the melting temperature (T_m_) of PPS@BC nanocomposites also revealed the same trend of increases in the values. The maximum achievable value was 284 °C for nanocomposite having 6% filler content. The graphic evaluation of T_m_ and T_c_ for PPS and PPS@BC nanocomposites is presented in Figure 10.

From the DSC trace of the cooling cycle, it is evident that the crystallization process of the PPS was accelerated by the addition of the filler, as there was a narrowing of the crystallization peak. The degree of supercooling (∆T) in the Figure 11 shows a decrease in value compared with that required for the neat PPS. So, crystallization in the nanocomposites is higher as compared to that required for neat PPS. The drop in value of the degree of supercooling indicated that the nucleation process was enhanced due to the presence of the BC nanofiller.

The degree of crystallization showed a continuous decrease of up to 16% with the addition of the filler. The glass transition temperature showed an initial decline in value until PPS-6 and after this there isrise in the parameter until PPS-10. The glass transition curves are presented in Figure 12. As it is easier for the nanocomposites with lower glass transition values to undergo molding, only 6% doping with the filler was found to be feasible for the development of a nanocomposite. This was accomplished with a 17 °C lower glass transition temperature than PPS. The comparative evaluation of the degree of crystallinity and glass transition temperatures for each sample is presented in Figure 13.

The PPS@BC nanocomposite blends showed two separate peaks for the melting and crystallization processes. The heat of crystallization for the PPS@BC nanocomposites was observed to increase continuously with an increase in the dopant content. This indicated the supportive role of BC in the crystallization step during mixing via coagulation. The temperature of the onset of crystallization for the PPS moved to a higher value on PPS@BC nanocomposites, whereas the temperature of onset for melting showed a comparatively smaller rise in value, as presented in Figure 10. The rise in temperature of the crystallization of PPS predicts improved nucleation with the blending of the prepared BC. It was evidenced that a 6% addition of BC maximally contributed to the increase in the crystallization rate.

### 4.7. Surface Morphology and Composition of PPS@BC

The FESEM imaging results of the PPS@BC nanocomposite and neat PPS, using various resolutions, are shown in Figure 14. All images reveal less agglomeration, good blending, and the uniform dispersion of BNC in the PPS matrix. These parameters are required for improvements in the mechanical properties (such as fracture resistance, tensile strength, and elongation at the break) of nanocomposites. A morphology of the neat resin of PPS was noted, with an elongated crack appearing on the bulky structure. Nanocomposites revealed the conversion of the PPS morphology into a fine-grained structure with reduced surface cracks and roughness. This ensured the productive interaction of synthesized BC into the polymer during coagulation. The filler used as a dopant was observed to have good penetration into the inter-spherulitic and interlamellar regions of the PPS chains [53,54]. The diffusion of BC remained homogenous even after an increase in the dopant concentration. This rapid diffusion is possibly due to chaos in the crystallite structure during the coagulation process. During composite formation, the change in the morphology of the inner structure of the polymer skeleton, with the creation of voids throughout the cross-section, is noticed.

There are distinguishing features, such as interior voids and the continuous saturation of BC nanoparticles into the polymeric matrix. No evidence of the aggregation of BC was perceived, indicating a supportive role of the coagulation method in the wetting of the PPS resin with filler. Carbon-based fillers have been reported to increase the porosity of polymer nanocomposites due to higher surface areas [53]. The detailed morphological images, obtained using EDS, are provided as BSE (backscattered electron) images in Figure 15. These in-depth images also correlate, in surface morphology, with FESEM images. The results of the weight fraction of carbon in the prepared nanocomposites are provided in Figure 16. This also indicated an increase in carbon content with the addition of BC.

## 5. Conclusions

The mesoporous BC, synthesized using the facile carbonization step, has a surface area and crystallinity that makes it effective for use as a filler. Its effective dispersion into the PPS@BC nanocomposites was achieved using a simple coagulation technique. The BC demonstrated a role in increasing the thermal stability of the nanocomposites. The crystallization process of the PPS nanocomposites was boosted, and the flexibility of the polymer—due to the decline in crystallinity—was improved when related to neat PPS. In polymer nanocomposites, the filler is known to control the morphology, crystallization, and nucleation processes. The polymer nanocomposites showed improved crystallinity and glass transition temperature values compared with the PPS resin. The appearance of a rubbery trend in the glass transition peak, with a decline in value of up to 6% of the filler mixing, is indicative of easy molding due to the introduction of the amorphous character. The melting temperature and crystallization temperature were slightly affected by the change in filler content. A rise in the heat of crystallization in the PPS@BC nanocomposites indicates ease in the crystallization process as a result of blending through coagulation. With the increase in the percentage of BC beyond 6%, the crystallization of the PPS skeleton was disturbed and a decrease in the crystallinity index was noted. The rise in amorphous character was directly associated with easy crosslinking of the polymer chain. The melting temperature and crystallization temperature were slightly affected by the change in filler content. A rise in the heat of crystallization in the PPS@BC nanocomposites indicates ease in the crystallization process as a result of blending through coagulation. A rise in the glass transition temperature after the lowest glass transition value in the sample with 6% filler loading was noticed. The 6% BC loading is indicative of the fact that excessive filler quantity has hindered the effective overlapping of the polymer chains. These properties predict the usage of synthesized nanocomposites in chemical processing industries for surface applications involving the coating of thread guides, molds, casings, driers, and valves.

## Figures and Tables

**Figure 1 polymers-15-01851-f001:**
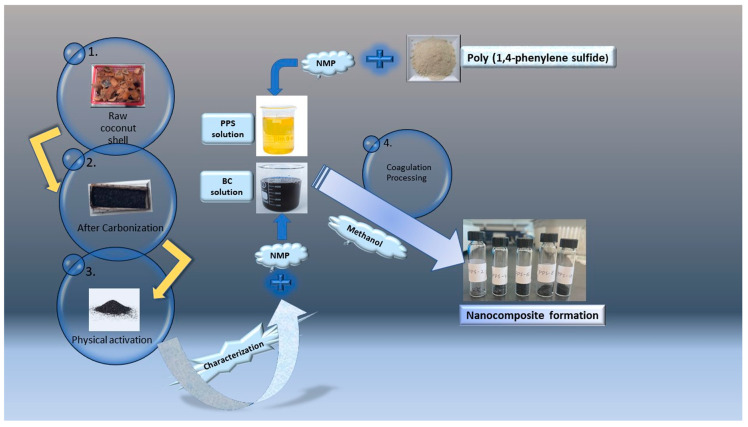
Schematic flow of the experimental workup.

**Figure 2 polymers-15-01851-f002:**
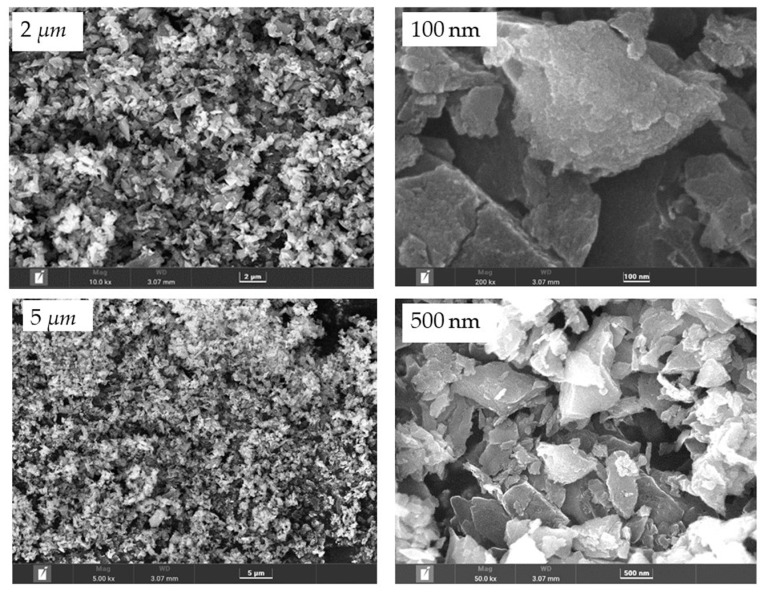
FESEM overview of BC at different magnifications.

**Figure 3 polymers-15-01851-f003:**
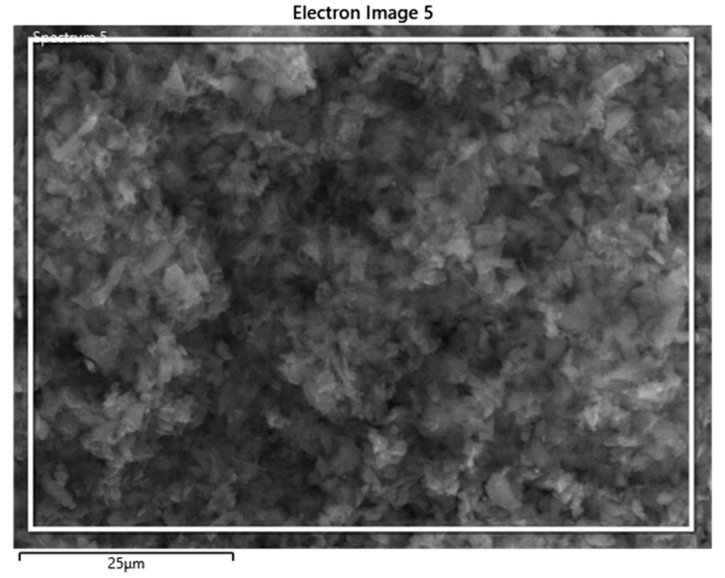
BSE image of BC.

**Figure 4 polymers-15-01851-f004:**
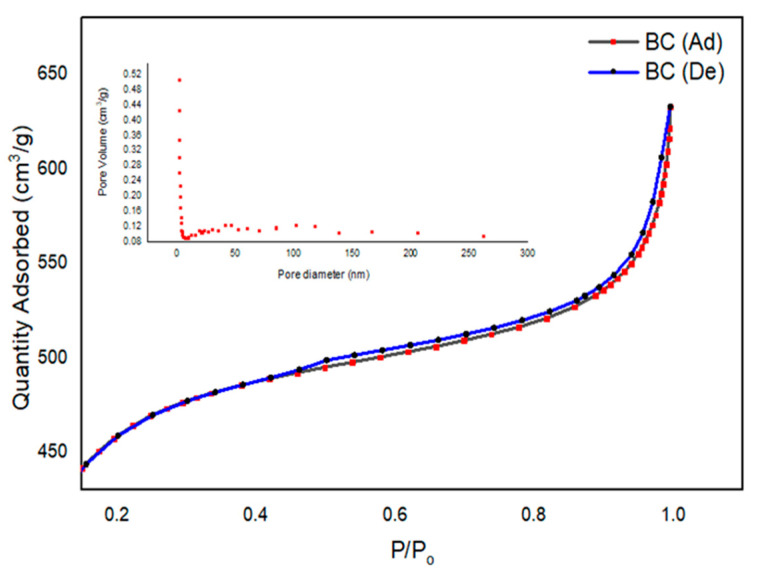
Isotherm linear plots and size distribution analysis of BC at STP.

**Figure 5 polymers-15-01851-f005:**
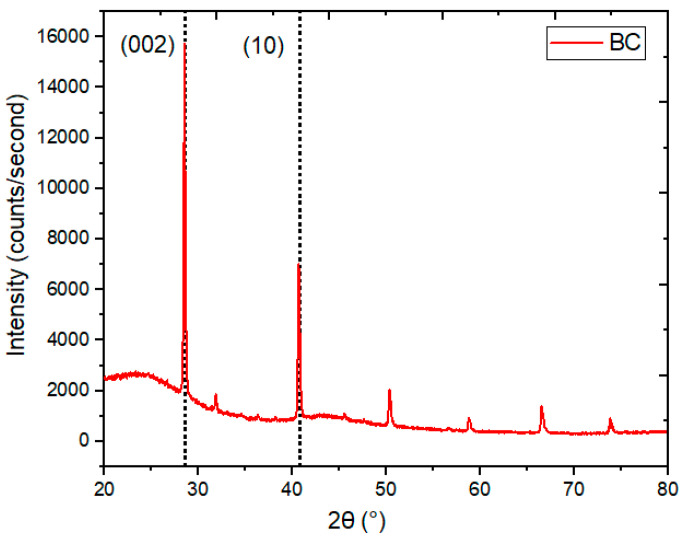
XRD spectral pattern of BC.

**Figure 6 polymers-15-01851-f006:**
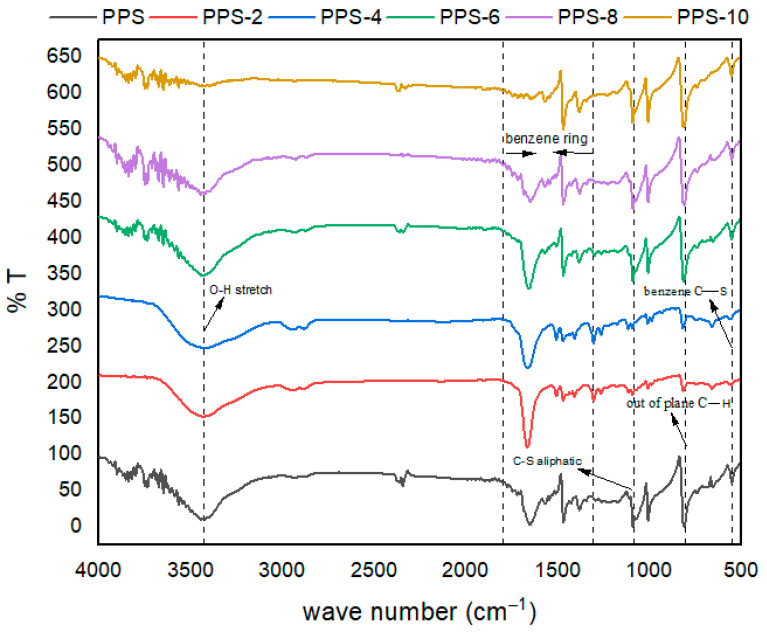
FTIR spectra of PPS@BC nanocomposites.

**Figure 7 polymers-15-01851-f007:**
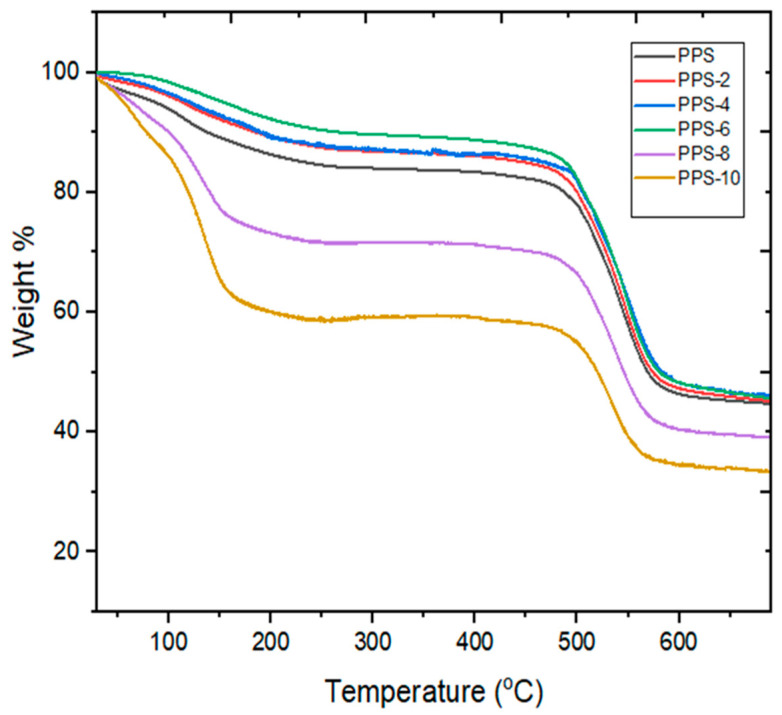
TGA curves of PPS@BC nanocomposites.

**Figure 8 polymers-15-01851-f008:**
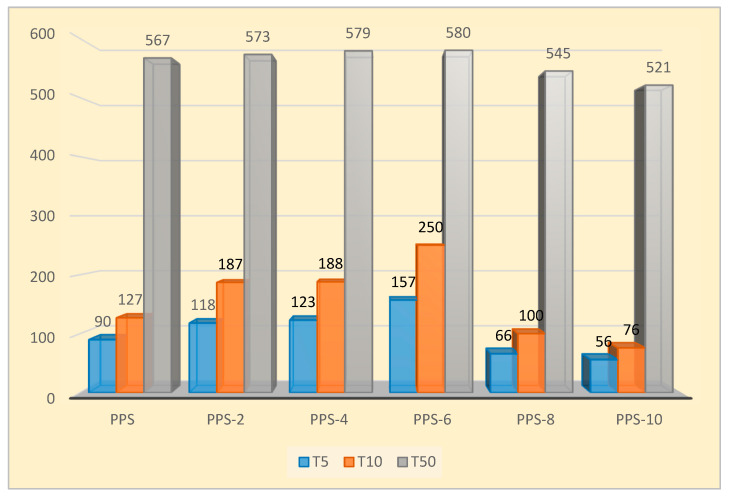
Comparative analysis T_5_, T_10_, and T_50_ PPS@BC nanocomposites.

**Figure 9 polymers-15-01851-f009:**
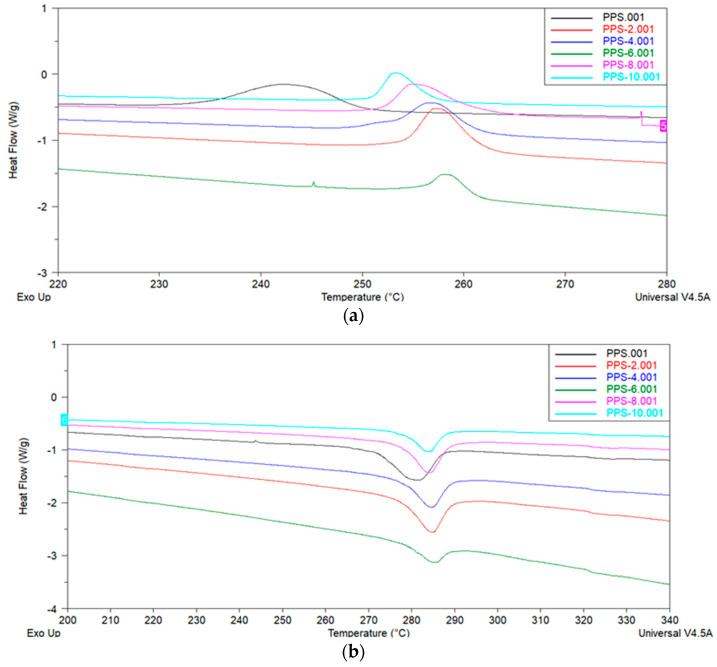
DSC curves of PPS@BC nanocomposites: (**a**) cooling cycle (**b**) heating cycle.

**Figure 10 polymers-15-01851-f010:**
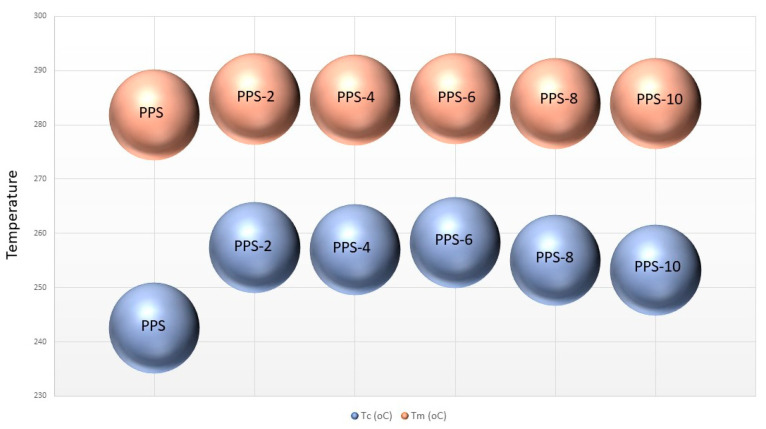
Comparative illustration of T_c_ and T_m_ of PPS@BC nanocomposites.

**Figure 11 polymers-15-01851-f011:**
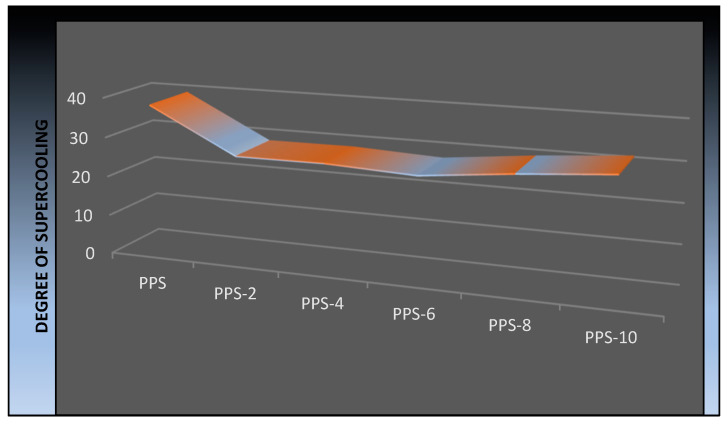
Graphic illustration of the change in the degree of supercooling of the PPS@BC nanocomposites.

**Figure 12 polymers-15-01851-f012:**
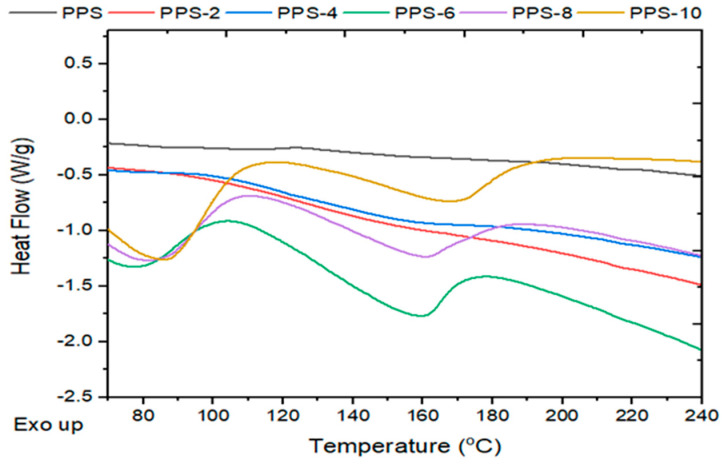
Glass transition analysis of the PPS@BC nanocomposites.

**Figure 13 polymers-15-01851-f013:**
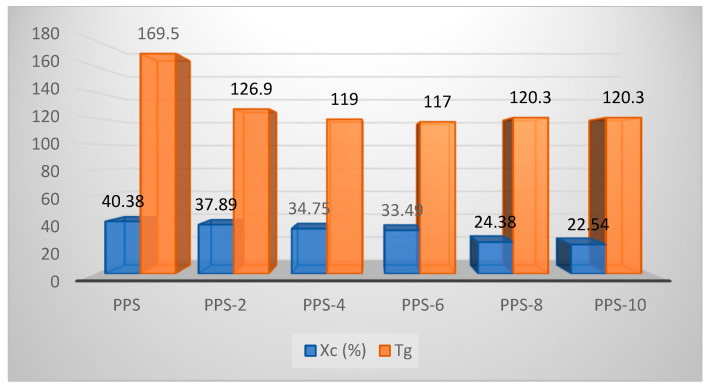
Comparison of the degree of crystallinity and glass transition temperatures of PPS@BC nanocomposites.

**Figure 14 polymers-15-01851-f014:**
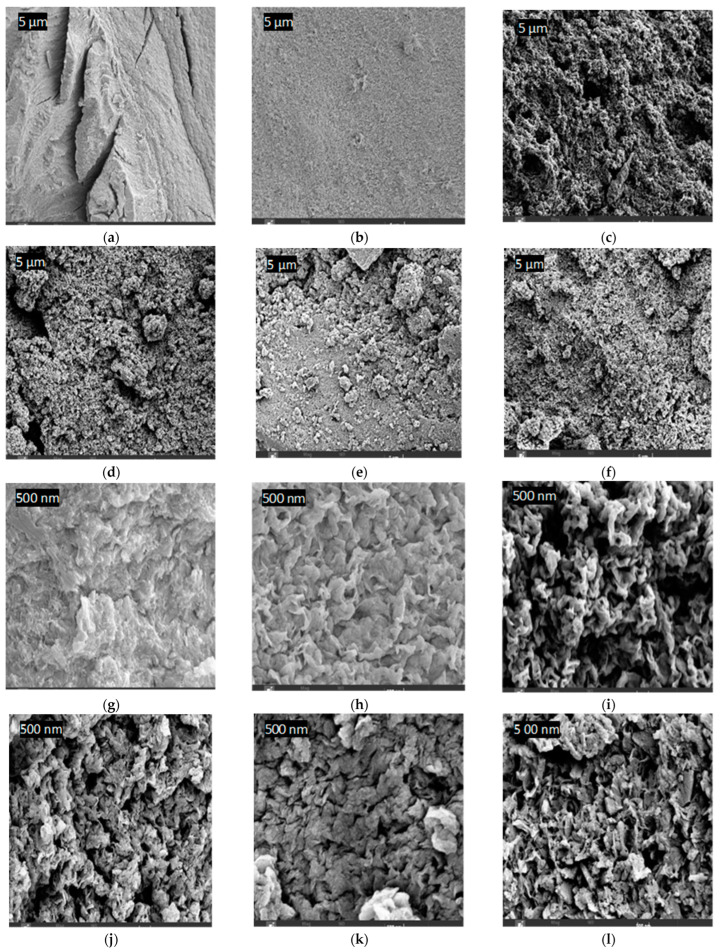
FESEM images of the samples: (**a**) PPS at 5 µm, (**b**) PPS-2 at 5 µm, (**c**) PPS-4 at 5 µm, (**d**) PPS-6 at 5 µm, (**e**) PPS-8 at 5 µm, (**f**) PPS-10 at 5 µm, (**g**) PPS at 500 nm, (**h**) PPS-2 at 500 nm, (**i**) PPS-4 at 500 nm, (**j**) PPS-6 at 500 nm, (**k**) PPS-8 at 500 nm, and (**l**) PPS-10 at 500 nm.

**Figure 15 polymers-15-01851-f015:**
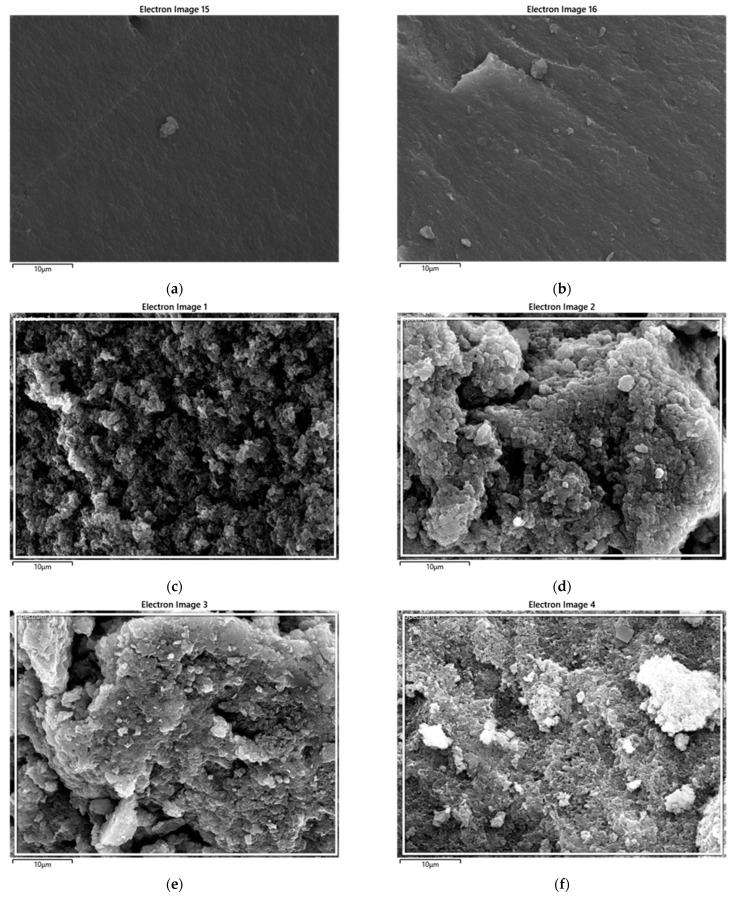
BSE images of PPS@BC nanocomposites: (**a**) PPS, (**b**) BSE PPS-2, (**c**) PPS-4, (**d**) PPS-6, (**e**) PPS-8, and (**f**) PPS-10.

**Figure 16 polymers-15-01851-f016:**
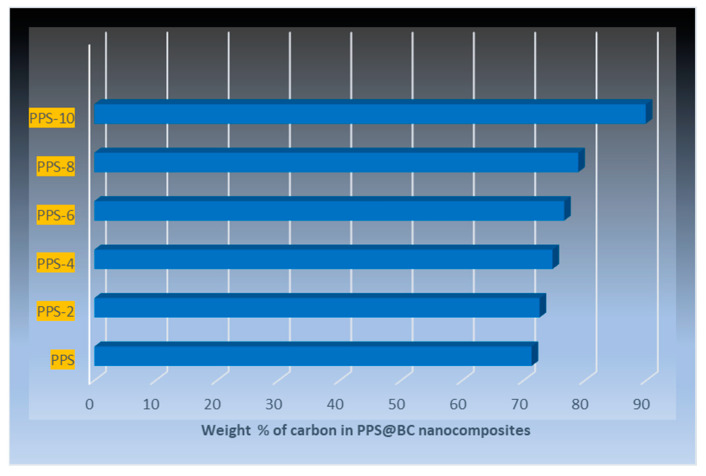
Quantitative analysis results of the weight fraction of carbon using EDS.

**Table 1 polymers-15-01851-t001:** EDS composition of BC.

Sample	Carbon (wt%)	Oxygen (wt%)	Potassium (wt%)
BC	90.2	9.5	0.3

**Table 2 polymers-15-01851-t002:** SAP parameters of BC.

Sample	Specific Surface Area (S_BET_)	Langmuir Surface Area (S_L_)	Average Pore Diameter (D)
BC	1517 m^2^/g	2175 m^2^/g	2.51 nm

**Table 3 polymers-15-01851-t003:** Peak position (°) of BC on the 2θ scale, as determined from the XRD pattern.

Sample	Plane (002)	Plane (10)
BC	28.4	40.7

**Table 4 polymers-15-01851-t004:** Structural information of BC, as determined from XRD patterns.

Sample	d_002_ (Å)	FWHM	L_c_ (nm)	L_a_ (nm)	N (Items)	δ (nm^−2^)	ε (×10^−3^)	ρ (g/cm^3^)	Crystallinity Index (%)
BC	3.140	0.195	42.024	90.585	13.383	0.566	3.362	0.243	53.077

**Table 5 polymers-15-01851-t005:** FTIR data of PPS@BC nanocomposites.

Samples	Wavenumber (cm^−1^)						
	O–H (Stret)	C–H (Stret)	Benz Ring (Sym Stretch)	Benz Ring (Asym Stret)	C–S (Aliph Stret)	C–H (Out-of-Plane)	Benz Ring (Sym Ring-s Stretch)
PPS	-	2373 (w)	1454 (s)	1382 (w)	1087 (w)	807 (m)	476 (m)
PPS-2	3433 (w)	2333 (w)	1647	1462 (w)	1082 (w)	811 (m)	469 (m)
PPS-4	3433 (s)	2365 (w)	1652 (s)	1468 (w)	1087 (w)	817 (m)	475 (m)
PPS-6	3433 (s)	2341 (w)	1657 (s)	1468 (w)	1089 (w)	817 (m)	480 (m)
PPS-8	3433 (s)	2398 (w)	1657 (s)	1468 (w)	1089 (w)	811 (m)	478 (m)
PPS-10	3433 (s)	2364 (w)	1652 (s)	1468 (w)	1084 (w)	811 (m)	475 (m)

Aliph, aliphatic; stret, stretching; sym, symmetric; asym, asymmetric; benz, benzene; w, weak; s, sharp; m, medium.

**Table 6 polymers-15-01851-t006:** TGA data of PPS@BC nanocomposites.

Compounds	T_5_ (°C)	T_10_ (°C)	T_50_ (°C)	T_f_ (°C)	R_f_ (%)
PPS	90	127	567	646	45
PPS-2	118	187	573	654	46
PPS-4	123	188	579	656	46
PPS-6	157	250	580	659	46
PPS-8	66	100	545	645	39
PPS-10	56	76	521	606	34

**Table 7 polymers-15-01851-t007:** Data obtained from the DSC trace of PPS@BC nanocomposites.

Compounds	T_oc_ (°C)	T_c_ (°C)	∆H_c_ (J/g)	T_om_ (°C)	T_m_ (°C)	∆H_m_ (J/g)	∆T (°C)	X_c_ (%)	T_g_ (°C)
PPS	252.1	242.5	45.2	261.7	281.8	35.9	38.5	40.4	134.7
PPS-2	265.2	257.4	41.6	271.6	284.8	29.2	27.4	37.89	129.6
PPS-4	264.8	256.9	35.8	263.4	284.5	28.6	27.5	34.7	119.4
PPS-6	269.5	258.3	36.0	274.5	284.8	12.0	26.5	33.5	117.5
PPS-8	268.4	255.1	25.7	265.6	283.9	28.6	28.8	24.4	135.4
PPS-10	269.5	253.2	22.7	268.8	283.9	16.7	30.6	22.5	149.8

## Data Availability

Not applicable.

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
