# Peer review of "Experimental Correlation of the Role of Synthesized Biochar on Thermal, Morphological, and Crystalline Properties of Coagulation Processed Poly(1,4-phenylene sulfide) Nanocomposites"

_polymers, 2023, doi:10.3390/polym15081851_

Round 1

Reviewer 1 Report

The paper reports a novel use of biochar in production of nanocomposites and will be of interest to researchers in the field. The authors may wish to check or comment on the following:

p. 4 line 164: Is the FESEM voltage range actually 5-200KV, maybe this is the magnification range used?

p. 4 line 170: probably need to detail BSE as backscatter imaging?

p. 5 fig 3 (b): Maybe a table would be sufficient?

p. 9 Table 5: Errors inthe TGA data could be helpful

p. 10 Fig. 8: Does there need to be a table and figure with the same data?

p. 11 Table 6: errors in the DSC temperatures and relevant significant figures should be reviewed.

p. 12 Figures 9 and 10: do the tables and figures showing the same data need to be repeated?

p. 16: Fig 15: Errors in calculated wt% may be helpful

Could the char fraction data using the current material be compared to more traditional materials in the literature for comparison?

Author Response

Pl find the report attached

Reviewer 2 Report

Nice work. 

Please provide qualitative results in the abstract section.

Include recent literatures and some significant papers. The following literature could be helpful "Strengthening mechanisms of graphene sheets in aluminium matrix nanocomposites. Materials & Design88, pp.983-989"; "A Magnetic Nanoparticle-Doped Photopolymer for Holographic Recording. Polymers14(9), p.1858"; "Recent trends in drilling of carbon fiber reinforced polymers (CFRPs): A state-of-the-art review. Journal of Manufacturing Processes69, pp.47-68".

The flow chart of the process could be helpful for the other researchers.

Improve the conclusion section.

Author Response

Pl find the report attached

Reviewer 3 Report

please see the attached document

Author Response

Pl find the report attached
